# Modeling the Phase Equilibria of Associating Polymers in Porous Media with Respect to Chromatographic Applications

**DOI:** 10.3390/polym14153182

**Published:** 2022-08-04

**Authors:** Xiu Wang, Zuzana Limpouchová, Karel Procházka, Rahul Kumar Raya, Yonggang Min

**Affiliations:** 1School of Materials and Energy, Guangdong University of Technology, Guangzhou 510006, China; 2Department of Physical and Macromolecular Chemistry, Faculty of Science, Charles University, Hlavova 8, 128 43 Prague , Czech Republic

**Keywords:** Monte Carlo simulation, amphiphilic diblock copolymer, association, partition coefficient, critical micelle concentration, size-exclusion chromatography, micellar liquid chromatography

## Abstract

Associating copolymers self-assemble during their passage through a liquid chromatography (LC) column, and the elution differs from that of common non-associating polymers. This computational study aims at elucidating the mechanism of their unique and intricate chromatographic behavior. We focused on amphiphilic diblock copolymers in selective solvents, performed the Monte Carlo (MC) simulations of their partitioning between a bulk solvent (mobile phase) and a cylindrical pore (stationary phase), and investigated the concentration dependences of the partition coefficient and of other functions describing the phase behavior. The observed abruptly changing concentration dependences of the effective partition coefficient demonstrate the significant impact of the association of copolymers with their partitioning between the two phases. The performed simulations reveal the intricate interplay of the entropy-driven and the enthalpy-driven processes, elucidate at the molecular level how the self-assembly affects the chromatographic behavior, and provide useful hints for the analysis of experimental elution curves of associating polymers.

## 1. Introduction

Modern liquid chromatography (LC) techniques, including size-exclusion chromatography (SEC), interaction chromatography (IC), and other related chromatography variants, are the currently used benchmark approaches for the separation, purification, and analysis of polymers [1,2,3,4,5,6]. Conventional chromatographic analysis assumes that polymeric analytes dissolve in the mobile phase, interact with the porous stationary phase, and finally elute from the LC column at distinct elution volumes depending on the molar mass, chemical composition, and chain architecture. This regular separation (or analysis) is usually performed in a fairly dilute regime, which, to a large extent, prevents the association of polymer chains and consequent complications and minimizes the risk of erroneous results [7]. However, in some special cases, the aggregation of some compounds in chromatographic columns is unavoidable or even desirable; therefore, it is worthy of being further investigated, understood, and exploited. For instance, micellar liquid chromatography (MLC) employs the surfactant-based mobile phase above the critical micelle concentration (CMC) and capitalizes on the fact that the intricate interactions of analytes with micelles formed in the mobile phase and interactions with surfactants adsorbed in the stationary phase give rise to unique partitioning behavior [8]. If non-toxic biocompatible micellar media are used, MLC can be exploited as a robust green characterization technique for a broad range of biomedically relevant analytes [9,10,11,12,13,14,15].

Non-ionic associating polymers with a relatively simple molecular structure, such as amphiphilic diblock copolymers, undergo a one-step concentration-dependent association process that obeys the closed association scheme [16] and is reminiscent of the micellization of surfactants. Below the CMC, the polymer chains do not associate. At the CMC, the associates with a relatively narrow distribution of association numbers start to form. When the concentration of solution exceeds the CMC, the concentration of non-associated chains (unimers) remains equal to the CMC, and only the concentration of micelles, the mass and size of which do not change, increases with further increase in polymer concentration. The phase behavior of associating polymers passing through an LC column can be described by the following equilibria (in this article, we use bulk and pore to represent the mobile phase and the stationary phase, respectively): (1)$$m_{\mathrm{bulk}}P_{\mathrm{bulk}}{}_{\overset{\leftarrow }{k_{d,\mathrm{bulk}}}}^{\underset{\to }{k_{a,\mathrm{bulk}}}}P_{m,\mathrm{bulk}}$$
(2)$$P_{\mathrm{bulk}}{}_{\overset{\leftarrow }{\;}}^{\underset{\to }{K_{p}}}P_{\mathrm{pore}}$$
(3)$$m_{\mathrm{pore}}P_{\mathrm{pore}}{}_{\overset{\leftarrow }{k_{d,\mathrm{pore}}}}^{\underset{\to }{k_{a,\mathrm{pore}}}}P_{m,\mathrm{pore}}$$
where P represents the free chains; $P_{m}$ represents the aggregates (micelles) with the number-average association number, m; $K_{p}$ represents the partition coefficient of free chains partitioning between two phases; and $k_{a}$ and $k_{d}$ stand for the association and dissociation rate constants, respectively. The model assuming the above equilibria (Equations (Equation 1)–(Equation 3)) is based on the following working hypothesis. The studied associating polymers form quite large spherical core-shell associates in the bulk solvent (mobile phase), which do not enter relatively narrow pores. The non-associated chains (unimers) can enter the pores, and the phase equilibrium is characterized by the partition coefficient, $K_{p}$. Above the CMC, the concentration of unimers in the pores is $K_{p}\times \mathrm{CMC}$, i.e., low in the SEC regime, but it can be fairly high in the IC regime. The confined chains in the pores can, in principle, form spherical associates with lower association numbers than those in the bulk, or associates differing in morphology. The CMC in the pores and in the bulk can differ slightly due, in part, to steric confinement and to important interactions between polymers and pore walls. Hence, the association in pores, which would suck free chains into the pores and significantly affect the equilibria, cannot be *a priori* precluded, particularly in the case of IC. Our simulation study aims at the proof or refutation of individual processes involved in the above hypothesis.

Over the last 30 years, both basic and application-oriented research on LC have been performed extensively [3,4,5,6], but only several experimental and a few theoretical papers on the chromatographic separation and analysis of reversibly associating systems have been published [17,18,19,20,21,22]. For instance, in the recent seminal work, Adawy and Groves monitored the protein aggregation using SEC [23]. Despite these achievements, information in this realm is still limited. In order to fully explore the application potential of LC in the research of associating polymer systems, it is desirable to study the interplay of entropic and enthalpic effects comprising the polymer self-assembly in bulk solutions, sterically confined assembly in pores, and polymer partitioning affected by the competition of steric exclusion and adsorption on the pore walls. The computer simulation is an intelligent, powerful, and extraordinarily suitable approach for the investigation of the above-listed interrelated processes. Although the bulk-pore (also called the twin-box) model has been well developed and extensively employed in simulations of the partitioning of various polymers [24,25,26,27,28,29,30,31,32,33,34,35,36,37,38], to the best of our knowledge, the computer-aided interpretation of LC data on the partitioning of associating polymers has not yet been published. In this work, we study the bulk association of block copolymers in a selective solvent, and their partitioning and adsorption on porous media. In contrast to numerous papers that focus on the self-assembling phenomena of associating polymers and particularly on the morphology of aggregates under confinement [39,40,41,42,43,44,45,46,47,48,49,50], we pay special attention to the impact of self-assembly (micellization) on the phase equilibria and on the retention behavior in both the SEC and IC regimes. From the theoretical viewpoint, we are looking for universal features of the chromatographic behavior of associating systems such as amphiphilic copolymers, surfactants, proteins, and other biomolecules. From the practical point of view, we focus on the possibility (i) to separate the self-assembling copolymers differing in composition and (ii) to estimate their CMCs. We believe that the knowledge obtained could contribute to the development of chromatographic techniques (including MLC) and to the optimization of experimental conditions of chromatographic studies of self-assembling systems.

## 2. Methods and Simulation Details

A simulated non-ionic amphiphilic AB diblock copolymer chain consists of a solvophilic (soluble) block (A) and a solvophobic (insoluble) block (B), and the chain length is fixed at $N=N_{A}+N_{B}=64$. The length of the solvophobic block, $N_{B}$, varies from 16 to 40, and the length of block A , $N_{A}$, varies accordingly from 48 to 24. All chains are simulated on a cubic lattice. Each lattice site is occupied by either a solvent molecule (S) or by a monomer (A or B), except the sites excluded by the pore wall. Note that this exclusion has nothing in common with the entropic depletion effect, i.e., with the decrease in concentration of polymer chains in the vicinity of impermeable inert walls due to the reduced number of possible chain conformations close to the wall [31]. This entropic effect plays a non-negligible role in the studied systems and appreciably lowers the concentration of polymer beads close to the inert wall, but as the cylindrical wall generally passes between the lattice points, the distance of some lattice points from the wall is r<1, and these positions are excluded by the definition of the interacting potential (Equation (Equation 5)). We apply the single-site bond fluctuation model, which generates 26 possible bonds (connecting the adjacent monomers) fluctuating between 1 and $\sqrt{3}$ by the permutation of the vectors of (±1, 0, 0), (±1, ±1, 0) and (±1, ±1, ±1) [51]. Note that neither the overlap of polymer beads (when two or more monomers occupy the same lattice site) nor the intersection of bonds is permitted [52,53]. A bead (monomer or solvent) interacts with its neighbors in the range of distance, r, from 1 to $\sqrt{3}$, and the pairwise potential energy, $\varepsilon_{\alpha \beta }\left(r\right)$, is defined as
(4)$$\varepsilon_{\alpha \beta }\left(r\right)=\left\{\begin{array}{cc} \infty & \left(r<1\right)\\ \epsilon_{\alpha \beta } & \left(1\leq r\leq \sqrt{3}\right)\\ 0 & \left(r>\sqrt{3}\right) \end{array}\right.,$$
where α and β stand for the solvent (S) or the monomers (A and B), and $\epsilon_{\alpha \beta }$ represents the interaction strength. The bulk mobile phase is modeled as an unconfined cube with the dimensions of 100×100×100 ($L_{x,\mathrm{bulk}}\times L_{y,\mathrm{bulk}}\times L_{z,\mathrm{bulk}}$, in lattice units), and the pore (stationary phase) is modeled as an impermeable cylindrical tube with a variable diameter, D. The schematic representation of the employed bulk-pore model is shown in Appendix A. For a given pore diameter, D, we change $L_{x,\mathrm{pore}}$ to keep the volume ratio $V_{\mathrm{pore}}/V_{\mathrm{bulk}}=1$. For both the bulk and the pore, periodic boundary conditions (PBC) are imposed in all directions except the impermeable shell of the cylindrical pore. The pore is filled with solvent molecules, and the copolymer chains entering the pore interact with the wall, i.e., with the inner surface of the pore. The bead–wall interaction, $\varepsilon_{W\alpha }\left(r\right)$, is defined as
(5)$$\varepsilon_{W\alpha }\left(r\right)=\left\{\begin{array}{cc} \infty & \left(r<1\right)\\ \epsilon_{W\alpha } & \left(1\leq r\leq \sqrt{2}\right)\\ 0 & \left(r>\sqrt{2}\right) \end{array}\right.$$

Here, the subscript α represents A or B or S; the subscript W stands for the pore wall; and r is the normal distance from the bead to the concave inner surface. Note that $\epsilon_{W\alpha }=0$ and $\epsilon_{W\alpha }<0$ stand for an inert wall and for an attractive wall interacting with species α, respectively. This employed square potential implies that neither monomers nor solvent molecules can occupy the lattice sites of r<1. In this simulation study, the values of all interaction parameters are expressed in the energy units, $k_{b}T_{0}$, where $k_{b}$ is the Boltzmann constant and $T_{0}$ stands for the temperature of reference state.

We use the modified dynamic configuration-bias Monte Carlo (CBMC) algorithm based on the method originally developed by Siepmann and Frenkel [54]. The original CBMC employs the implicit solvent, which means that each unoccupied lattice position implicitly contains a solvent bead. Simultaneously, all interaction parameters with solvent are by definition zero, i.e., $\epsilon_{\alpha S}=0$, and the nonzero interactions between polymer beads, $\epsilon_{\alpha \beta }$ (α,β≠S), indirectly model the solvent quality [55,56,57,58]. The choice of zero interaction parameters with solvent simplifies the acceptance criterion due to the fact that the transition probability between two states can be expressed as the ratio of the Boltzmann factor and the Rosenbluth weight reflecting the interaction energy [54].

The CBMC method has to be modified when applied to copolymers in selective solvents. As described in our earlier papers [59,60], we use the variant with tailored weights, reflecting the fact that the interactions of beads A and B with the solvent differ. In this case, the transition probability contains the Boltzmann factor of the generated polymer conformation, which reflects the energy difference between the trial and original self-avoiding walk (SAW), including the energy contribution caused by the exchange of polymer beads and solvent molecules (see Figure 1).

A detailed explanation of the weighting factors used together with the justification of the modified acceptance criterion can be found in the Appendix A. The application of the modified CBMC is formally analogical to that of the original method. We express the modified weights by the energy difference parameter $z_{\alpha \beta }=\varepsilon_{\alpha \beta }-\varepsilon_{S\beta }$, where S denotes the solvent and α, β denote A, B, and W (pore wall). Then, the energy difference between the two systems differing in the bead type in a given lattice position *i* can be expressed as the sum of all $z_{\alpha \beta }$ of its neighbors (as exemplified in Figure 1), i.e., $Z_{i}=\sum {\left(z_{\alpha \beta }\right)}_{i}$.

Contrary to the original CBMC method proposed by Siepmann and Frenkel [54], which models the solvent quality indirectly by one parameter only, our modified approach requires the setting of several interaction parameters, $\epsilon_{\alpha \beta }$, which are neither simply related to parameters used in the original CBMC nor to the Flory–Huggins interaction parameter [61]. They are loosely related to the Lennard–Jones (LJ) interaction parameters (to both homo- and cross-interactions) [62]. Nevertheless, as our model does not include the explicit solvent–solvent interactions, i.e., $\epsilon_{\mathrm{SS}}=0$, the used $\epsilon_{\alpha \beta }$ are not directly proportional to the LJ parameters, which describe the interactions in vacuum and are used in explicit solvent model systems. The parameters used in the modified CBMC have to be estimated independently on the basis of mapping the data onto the experimental association behavior. Note that $\epsilon_{\mathrm{AS}}=0$ still describes a good solvent condition analogously to the original CBMC calculation, but the value $\epsilon_{\mathrm{BS}}=0.1$ already models a poor solvent and hence the combination of the two values above, i.e., $\epsilon_{\mathrm{AS}}=0$ and $\epsilon_{\mathrm{BS}}=0.1$, provides the selective solvent. The low value $\epsilon_{\mathrm{BS}}=0.1$ reflects the fact that in the used variant of single-site bond fluctuation model, all 26 neighbors (up to the distance $r=\sqrt{3}$, see Equation (Equation 4)) interact equally with the segment/solvent in a given lattice position, and the effect of interactions is therefore more than 4 times stronger than that in common CBMC simulations on the simple cubic lattice, where the number of interacting neighbors is only 6. In this paper, we do not emulate the behavior of any particular copolymer-solvent system but focus on the general behavior of self-assembling copolymers in selective solvents. Because the simulations in bulk yield a reversible equilibrium of well-defined core-shell micelles with unimer chains, i.e., the study faithfully emulates the behavior of real micellizing systems and fulfills the basic prerequisite of the working hypothesis, we are of the opinion that the interaction parameters have been set appropriately.

As previously mentioned, for the simulated amphiphilic diblock copolymers, we set $\epsilon_{\mathrm{AS}}=0$ to model a good solvent condition for block A and $\epsilon_{\mathrm{BS}}=0.1$ to describe the insolubility of block B. We set $\epsilon_{\mathrm{AB}}=0.15$ to describe the interaction between the incompatible A and B monomers. The parameters of other pairwise interactions are $\epsilon_{\mathrm{AA}}=\epsilon_{\mathrm{BB}}=\epsilon_{\mathrm{SS}}=0$. For the polymer–wall interaction, we use $\epsilon_{\mathrm{WA}}=\epsilon_{\mathrm{WB}}=\epsilon_{\mathrm{WS}}=0$ to model the SEC mode, and we use negative values to describe the attractive interaction of polymer beads to the inner surface of the cylindrical pore. Besides the local move, i.e., the deletion and regrowth of chains in the same box, we employ the swap of polymer chains between two boxes (akin to the Gibbs ensemble Monte Carlo simulation [63]) to enable their partitioning between two phases. We always performed at least $5\times 10^{8}$ CBMC steps consisting of molecule swaps and local moves with the ratio of 1:1 for each simulation run, except for some special cases when the simulations needed to be prolonged. A simulation trajectory was divided into 20 blocks to estimate the standard deviations of computed quantities. Additionally, we tested each simulation starting from several different initial configurations to confirm the reliability of the results.

We assume that two copolymers belong to the same micelle if two insoluble beads (B) from different chains are at neighboring lattice sites. This association criterion was also proposed and used by other authors [58,64,65,66,67].

In addition to studies of the bulk-pore partitioning, we performed several simulations separately in the bulk and in the pore to calculate the excess chemical potential of diblock copolymers,
(6)$$\mu^{\mathrm{ex}}=-k_{b}T\ln\langle W_{N}\rangle /g^{N-1},$$
where T is temperature and the constant g represents the total number of interacting neighbors in the model used (g=26). Note that the ensemble average of the Rosenbluth weight of the generated ghost chain, $\langle W_{N}\rangle$, is normalized by the ideal gas part, $g^{N-1}$. We performed up to $1\times 10^{9}$ CBMC steps for these individual runs and computed $\mu^{\mathrm{ex}}$ every $10^{6}$ steps by generating $10^{5}$ ghost chains.

In this study, the effective volume of the pore, $V_{\mathrm{eff},\mathrm{pore}}$, is represented by the number of lattice sites in the pore that can be occupied by polymer or by solvent beads, i.e., the lattice sites of r≥1 (r is the distance from the wall). Obviously, the effective volume of the bulk can be expressed as $V_{\mathrm{eff},\mathrm{bulk}}=L_{x,\mathrm{bulk}}\times L_{y,\mathrm{bulk}}\times L_{z,\mathrm{bulk}}$. To discuss the partitioning of a multiphase system encompassing the equilibria described by Equations (Equation 1)–(Equation 3), we define the effective partition coefficient of copolymers, K, as
(7)$$K=\frac{C_{\mathrm{pore}}}{C_{\mathrm{bulk}}}=\frac{n_{\mathrm{pore}}N/V_{\mathrm{eff},\mathrm{pore}}}{n_{\mathrm{bulk}}N/V_{\mathrm{eff},\mathrm{bulk}}}=\frac{n_{\mathrm{pore}}V_{\mathrm{eff},\mathrm{bulk}}}{n_{\mathrm{bulk}}V_{\mathrm{eff},\mathrm{pore}}},$$
where $C_{\mathrm{pore}}$ and $C_{\mathrm{bulk}}$ represent the total equilibrium concentrations of copolymer beads in the pore and the bulk, respectively; $n_{\mathrm{pore}}$ and $n_{\mathrm{bulk}}$ represent the numbers of copolymer chains in the pore and the bulk, respectively; and the total chain length $N=N_{A}+N_{B}=64$. If the association of copolymers does not occur in either phase, apparently, $K=K_{p}$, where $K_{p}$ stands for the partition coefficient of non-associated chains (see Equation (Equation 2)). For brevity, in this article, we often use the short term “partition coefficient” for the effective partition coefficient, K. The majority of the simulations has been performed at the temperature $T/T_{0}=1.7$. At end of the paper, when we show and discuss the results of simulations in the temperature range $T/T_{0}=1.5$ to 1.8; the actual value is always given at the pertinent place.

## 3. Results and Discussion

First, we studied the partitioning of symmetric A32B32 diblock copolymers in moderately narrow pores with inert walls, of which the diameters, D, range from 15 to 30. The parameters were used emulate the conditions of the SEC regime under mild confinements because the coil-to-pore size ratio, $\lambda =2R_{g}/D$, ranges from 0.36 to 0.72 (the simulated radius of gyration, $R_{g}$, of the single chain in bulk is 5.42). The simulations started with all the chains in the bulk corresponding to the injection of the polymer solution into the column in practical chromatography. Figure 2a depicts the variation of the effective partition coefficient, K, with the total concentration of beads, $C=\left(n_{\mathrm{bulk}}+n_{\mathrm{pore}}\right)N/\left(V_{\mathrm{eff},\mathrm{bulk}}+V_{\mathrm{eff},\mathrm{pore}}\right)$, ranging from $3.2\times 10^{-5}$ to $8.9\times 10^{-3}$. We first describe the data for D=15. In the dilute regime for $C\leq 1.5\times 10^{-3}$, K is approximately 0.2 and almost independent of C (decreases only negligibly with C), indicating that the effective solvent quality for the whole copolymer is slightly worse than the θ-solvent condition [34]. However, when the concentration is higher than $1.5\times 10^{-3}$, K drops steeply to 0.07 and then decreases slowly with increasing C. The trends of K vs. C are similar for all studied D. The values of K increase with increasing D for all concentrations in accordance with the basic feature of SEC [68]. For the widest pore of D=30, K starts at 0.68 (for $C=3.2\times 10^{-5}$) and decreases slightly with increasing C. As in other pore diameters, a sudden abrupt drop to 0.24 occurs when C reaches approximately $2.4\times 10^{-3}$, and afterwards, K decreases slightly with increasing C. In conclusion, the concentration dependence of associating diblock copolymers exhibits a pronounced decreasing sigmoidal shape and differs from the slightly increasing or decreasing (almost linear) curves observed for non-association polymers in current solvents differing in thermodynamic quality until the saturation of pores [34,35,36,37]. We performed another set of simulations starting with all polymer chains in the pore and obtained the same results (see Appendix A), which confirms (i) the ergodicity of the simulation procedure used and (ii) the data describing the equilibrium behavior.

The snapshots in Figure 2a corroborate our hypothesis that the observed steeply changing behavior with the sudden drop in the K vs. C curves is caused by the association of A32B32 copolymers in the bulk phase. At low concentrations, all chains are well dispersed in bulk, and the bulk-pore partitioning of unimers (Equation (Equation 2)) controls the phase equilibria. This behavior is analogous to that of non-associating systems. When the concentration exceeds the CMC, the unimers start to associate and the aggregates, i.e., the micelles with solvophobic cores and solvophilic shells, form in the mobile phase and coexist in equilibrium with unimers. The concentration of micelles increases fast with the increase in the total concentration, and at C>CMC, the phase equilibria and consequently the chromatographic behavior are strongly influenced by the micellization of copolymers in the mobile phase. The CMC value can be estimated from the intersection point of the extrapolated dotted curves: (i) the relatively flat part of the curve for low C (green) corresponding to dilute solutions containing only unimers; and (ii) the decreasing part (brown). The intersection at $C_{M}^{*}\approx 1.9\times 10^{-3}$ in Figure 2a indicates the first appearance of micelles, and the narrow region close to CMC corresponds to the metastable regime, in which the micelles start to form. In this region, the fluctuations grow exponentially for C→CMC [69], and thus the error bars in K vs. C are remarkably large. The steeply decreasing part reflects the initial rise in micelle concentration, while the ratio of unimer-to-micelle concentrations is still important, and the micelles do not yet dominate the partitioning. The dominance of micelles in bulk translates in the third almost flat (only slightly decreasing) low K part at high C. Note that the value of the directly estimated critical concentration, $C_{M}^{*}$, differs from the CMC of the bulk solution because the C axis measures the total concentration of copolymers in two boxes (two phases) and the association takes place only in bulk. Accordingly, the true CMC of the bulk solution can be estimated as $\mathrm{CMC}=C_{M}^{*}\left(V_{\mathrm{eff},\mathrm{bulk}}+V_{\mathrm{eff},\mathrm{pore}}\right)/V_{\mathrm{eff},\mathrm{bulk}}$. As the relationship between CMC and $C_{M}^{*}$ is straightforward, for simplicity, we will use only the generic term CMC in the remaining part of the paper when discussing the features and trends of the partitioning and their impacts on chromatographic behavior. Furthermore, the snapshots of copolymer chains inside the pores of D=15 and 30 corresponding to the partitioning at the highest total concentration, i.e., $C=8.9\times 10^{-3}$, are shown in Figure 2b. No micelle is observed to form in both pores and this will be discussed later. A slightly surprising small drop on the curves at the concentration ca. $3.5\times 10^{-3}$ in Figure 2a will be explained in the next part.

Using the criterion for discerning aggregates from unimers (described in the methodology section), we subsequently measured the number of free (non-associated) chains, $n_{\mathrm{free}}$, in the bulk and the pore. As depicted in Figure 3a, $n_{\mathrm{free}}$ in the bulk phase first increases (at the low concentrations), peaks at the point representing the CMC, then decreases slightly with increasing C. This behavior, which occurs in all pores differing in D, is a consequence of the closed association mechanism [16] and is consistent with the light scattering data [71,72] and with observations concerning the “anomalous micellization” [73]. Just below and immediately above the CMC, various temporary (irregular and strongly fluctuating) diffuse aggregates form, but with the increasing concentration, the regular micellization soon prevails over the formation of metastable aggregates. The concentration of free chains also fluctuates, and we observe an increase and peaking of unimer concentration in this metastable region, presumably as a result of (i) the chaotic and strongly fluctuating micellization dynamics and (ii) the fact that the distances between beads of different chains in loose metastable aggregates are larger than those in regular micelles, and some chains (which change their conformations rapidly) could not have been identified as parts of loose aggregates by the criterion based on a fixed distance between two pairs of insoluble beads from different chains. The dependence of $n_{\mathrm{free}}$ on C in the pore is analogous to that in the bulk but exhibits a strong pore size effect, i.e., $n_{\mathrm{free}}$ increases with increasing pore diameter, D, for a given C because of the weakening constraint exerted by the pore (Figure 3b).

To compare the information of bulk CMC provided by the bulk-pore partitioning simulations (or characterized by LC) with that obtained from bulk solutions only, i.e., without the bulk-pore partitioning, we plot $n_{\mathrm{free}}$ against the $C_{\mathrm{bulk}}$ for both the bulk solutions and the bulk phases of partitioning simulations in Appendix A. Note that the concentration of beads in the bulk phase is derived from $C_{\mathrm{bulk}}=C\left(V_{\mathrm{eff},\mathrm{pore}}+V_{\mathrm{eff},\mathrm{bulk}}\right)/\left(KV_{\mathrm{eff},\mathrm{pore}}+V_{\mathrm{eff},\mathrm{bulk}}\right)$. We also plot the partition coefficient, K, against $C_{\mathrm{bulk}}$ in Appendix A. The curves in Appendix A indicate that the CMCs provided by the bulk-pore partitioning simulations are consistent with that computed from the bulk solutions, and that the properties of associating polymer solutions can be chromatographically characterized.

We believe that the second drop on the curves in Figure 2a at the concentration ca. $3.5\times 10^{-3}$ and the peaking of unimer concentrations above the CMC in Figure 3 are results of fluctuations in the metastable region where the micelles start to form. When discussing the strong effect of fluctuations in the metastable region close to the CMC on the data presented in Figure 2 and Figure 3, it is worth mentioning that the shapes of K vs. C curves obtained in our study are very similar to those of $1/M_{w}$ vs. C curves measured by static light scattering [71,72], but our data, which are much more sensitive to fluctuations than the SLS results, show the second (small and relatively gradual) drop, which indicates the termination of the chaotic metastable regime. Note that the SLS data, which show the reciprocal values of the weight-average molar masses, are strongly affected by high masses of associates and almost ignore the fluctuating contribution of unimer chains. Hence, our study reveals that the transient regime spans from the CMC to ca. 2 × CMC and cannot be detected by experimental methods that monitor the number-average, weight-average, or z-average molar masses or sizes.

Further, we focus on the bulk-pore partitioning of free (non-associated) unimer chains. Figure 4a depicts the dependences of the partition coefficient of free unimers, $K_{p}=C_{\mathrm{free},\mathrm{pore}}/C_{\mathrm{free},\mathrm{bulk}}$, on the total concentration, C, for the systems with non-adsorbing walls, where $C_{\mathrm{free},\mathrm{pore}}$ and $C_{\mathrm{free},\mathrm{bulk}}$ represent the concentrations of free unimers in the pore and in the bulk, respectively. The values of $K_{p}$ are not constant even in the range of the lowest concentrations, and the positive slopes of curves grow with the increase in the pore diameter, D. The observed trend is opposite to that of a common non-associating species partitioning between the bulk and pore, of which $K_{p}$ is almost constant at low concentrations; then, it slightly increases or decreases with C depending on the solvent quality [35] and drops finally quite fast due to the saturation of the pore. In the studied system, the unimer concentrations in both the bulk and the pore are low and do not exceed the CMC in the whole C region, and hence the saturation effect does not come in account. The fact that $K_{p}$ grows non-negligibly with increasing C, which means that $C_{\mathrm{free},\mathrm{pore}}$ increases in spite of the fact that $C_{\mathrm{free},\mathrm{bulk}}$ remains constant, is slightly surprising. Nevertheless, we believe that this unexpected behavior can be explained by the following arguments: in the pores with D ranging from 15 to 30, which are relatively narrow for micelles, the appreciable confinement effect hinders the formation of multimolecular micelles. The unimer chains in pores do not associate, and $C_{\mathrm{free},\mathrm{pore}}$ can thus exceed the bulk CMC. Because the interaction of the insoluble block with the inert pore wall is more convenient than that with the solvent, the unimer chains are slightly energetically driven into pores where the insoluble blocks concentrate close to the walls. This process (pseudo-adsorption of hydrophobic blocks on inert pore walls) is reminiscent of the formation of the unimolecular layer of surfactant molecules at the water–air interface, with the hydrophobic tails stretched towards the air to avoid the hostile aqueous medium [74,75].

Intuitively, one expects that the low unimer concentration in bulk should play the role of the stop-factor, preventing the penetration of chains into pores, but the micelles in bulk, which are in a reversible equilibrium with unimers, can dissociate and serve as a reservoir, providing the unimer chains for the process outlined above. Even though the surface-to-volume ratio decreases with D, the behavior at the low concentrations is reminiscent of the behavior of surfactants and depends only on the surface, i.e., it increases linearly with D. At the low concentrations, the copolymer chains bind to the surface analogously to surfactants that lower the water–air (or water-oil) interface tension. The maximum number of the surface-adsorbed chains is proportional to the surface and does not depend on the volume. Therefore, the slope of $K_{p}$ vs. C increases with D. The situation is different at high concentrations when the behavior depends on the surface-to-volume ratio, but here we discuss the values of $K_{p}$ at the extremely low concentrations.

Nonetheless, we still need to confirm whether the A32B32 diblock copolymers can aggregate in the pore at high concentrations or not. To do so, we plot the fraction of free chains, $\nu_{\mathrm{free}}=n_{\mathrm{free}}/n$, in both the bulk and the pore against the total concentration C, where n represents the total number of beads in the bulk or in the pore. As expected, the curves of $\nu_{\mathrm{free}}$ vs. C for the bulk phase shown in Figure 4b almost overlap, indicating that the self-assembly in bulk is only slightly affected by the phase equilibrium with pores and by pore size. For a given D, the $\nu_{\mathrm{free}}$ of the bulk copolymers starts at 1 and slightly decreases (to ca. 0.84) with increasing C in the dilute regime, which implies that at least 84% of the bulk copolymers dissolve as single chains before the sudden drop at the CMC. At C>CMC, $n_{\mathrm{free}}$ remains constant, but the $\nu_{\mathrm{free}}$ of bulk copolymers decreases steadily and gradually with increasing C (see Figure 4b) because more and more micelles are formed in the bulk. The $\nu_{\mathrm{free}}$ vs. C plot thus confirms the conclusion on self-assembly drawn from Figure 2. Figure 4b shows that the fractions, $\nu_{\mathrm{free}}$, in the pores of all diameters are almost constant (close to 1) in the whole concentration region but the values for D=30 are lower than those for D=15. This provides unambiguous proof that (i) the micelles formed in the bulk do not enter the pores; and (ii) the association of A32B32 copolymers in the narrowest pore is strongly sterically prohibited, but small aggregates with low association numbers can form under the moderate confinement, i.e., in the pore of D=30.

Using the criterion for discerning the associated from the non-associated chains, we assessed the number-average association number, $\bar{A_{s}}$. We plot $\bar{A_{s}}$ against the bulk concentration, $C_{\mathrm{bulk}}$, in Appendix A. The shape of the plot agrees with the conclusions drawn from the concentration dependences of the partition coefficient and from the plots of $n_{\mathrm{free}}$ vs. C. Additionally, the plots of the number distribution, $m_{n}\left(A_{s}\right)$, and weight distribution, $m_{w}\left(A_{s}\right)$, of the association numbers are shown in Appendix A. Here, we would like to note that the closed association scheme is a simplification of the behavior of real systems, and the same changes in $\bar{A_{s}}$ and in $n_{\mathrm{free}}$ for C>CMC were also reported by other experimentalists and computational scientists who studied the micellization in the bulk [59,76,77,78,79].

All data presented so far indicate that A32B32 copolymers can hardy associate inside the narrow and medium narrow cylindrical pores (D up to 30) with inert walls. The data presented in Appendix A corroborate this conclusion. They show that A32B32 do not associate even if they were initially inserted into the pores. The chains escape fast from the pore and associate in the bulk phase. Because the strong confinement effect obviously prevents the formation of micelles in narrow pores, we were curious about the behavior in wide pores. Hence, we enlarged the pore to D=60 and performed two sets of simulations at high concentrations, starting from both the bulk-located and the pore-located chain configurations. As shown in Figure 4c, the results of these two sets of simulations differ considerably, indicating the frozen non-equilibrium behavior in the case that all chains were initially in the pore. The unacceptably high K for SEC is apparently due to the fact that the chains associate in wide pores at concentrations above the CMC and the confined micelles are trapped there due to several contributing effects. Steric obstacles not only restrict the motion of crowded bulky micelles and prevent their escape into the bulk phase but also hinder their reorganization and dissociation. The escape of free chains from pores, which is expected to shift the equilibria in favor of the associates in bulk (see Equations (Equation 1)–(Equation 3)), is also inefficient because the transfer of unimers from pores into bulk controlled by low unimer concentrations and by tiny concentration gradient is extremely slow.

The simulations starting with all chains in the bulk faithfully emulate the equilibration process in SEC and IC: during the gradual passage through the column, the chains at the front edge of the analyte zone first enter the empty pores from the mobile phase, and hence the accumulation and stacking of chains in pores do not occur. In the real systems studied by experimental chromatography, as well as in model systems studied by our simulations, the associates in the bulk are relatively far from each other and the free chains can move and enter the pores without major obstacles. Therefore, we are persuaded that the simulations starting with chains in the bulk provide the equilibrium data.

To gain insight into the association of A32B32 diblock copolymers inside the pores, we performed the separate CBMC simulations in the pores of different diameters, D, excluding the possibility of the bulk-pore partitioning. In Figure 5a, we plot the fraction of free chains in the pore, $\nu_{\mathrm{free}}$, against D for the constant concentration $C_{\mathrm{pore}}=2.1\times 10^{-2}$, i.e., for 324 chains in the pore. In this case, when the concentration is high and the chains cannot escape from the pores, the association takes place even in the narrowest studied pore of D=20 ($\nu_{\mathrm{free}}$ is merely 0.32). The corresponding snapshot in Figure 5a shows that the associating copolymers under confinement do not form typical micelles like those in bulk, which agrees with extensive studies of other authors [40,46,48]. Therefore, we do not show the simulation data here and concentrate on the findings important for chromatography. Figure 5a shows that $\nu_{\mathrm{free}}$ decreases rapidly with increasing D and reaches a plateau (ca. 0.05) for D≥40. The extremely low $\nu_{\mathrm{free}}$ indicates significant association due to high $k_{a,\mathrm{pore}}$ (see Equation (Equation 3)). The snapshot for D=60 in Figure 5a reveals the micelle-like clusters in wide pores with solvophobic cores concentrated close to the pore center. In contrast, the snapshot for D=20 shows that the blocks (B) gather near the pore surface and the solvophilic ones concentrate near the pore center. To confirm these observations, we plot the concentration of beads, Φ, against the distance from the pore center perpendicular to the pore axis, r, for the low and high concentrations in Figure 5b,c. For all D, we see that at the extremely low concentration ($C_{\mathrm{pore}}=3.2\times 10^{-4}$), i.e., only 5 single chains in the pore, $\Phi_{A}$ is higher than $\Phi_{B}$ in the central region, and lower than $\Phi_{B}$ near the pore wall. At the high concentration ($C_{\mathrm{pore}}=2.1\times 10^{-2}$), $\Phi_{B}$ is considerably higher than $\Phi_{A}$ in the pore center and lower than $\Phi_{A}$ near the wall for D=40 and 60 as a result of the micelle formation. Nevertheless, for D=20, both curves of $\Phi_{A}$ and $\Phi_{B}$ vs. r are analogous to those at the low concentrations, which can be attributed to the strong confinement effect. Hence, the conclusions drawn from the concentration profiles agree with those drawn from the snapshots in Figure 5a.

From the simulations performed separately in the bulk solution and in the pores of various D, i.e., without taking the partitioning equilibrium in account, we evaluated the excess chemical potential, $\mu^{\mathrm{ex}}$, of A32B32 diblock copolymers using the Rosenbluth method as described in the methodology section [80,81]. In Figure 5d, we plot $\mu^{\mathrm{ex}}$ against the concentration of beads in the simulation box, C, for the copolymers in the bulk and in the pores of which the diameter, D, ranges from 20 to 60. The comparison of curves for the bulk phase and the pores differing in diameter is noteworthy. In the dilute regime, i.e., $C\leq 1.5\times 10^{-3}$ (see the inset of Figure 5d), the copolymers in the bulk acquire the highest values of $\mu^{\mathrm{ex}}$, and the $\mu^{\mathrm{ex}}$ of the confined chains decreases with decreasing D at a given concentration, C. This means that the A32B32 diblock copolymers in the narrowest pore of D=20 have the lowest $\mu^{\mathrm{ex}}$. Nonetheless, the difference in $\mu^{\mathrm{ex}}$ between the bulk and the pore, which controls the bulk-pore partitioning in the SEC mode at rather low concentrations (or at C<CMC), is small regardless of the pore diameter, D. Note that the shape of $\mu^{\mathrm{ex}}$ vs. C curves at higher concentrations is particularly interesting. The chemical potential, $\mu^{\mathrm{ex}}$, for the narrow pore of D=20 decreases smoothly with C in the whole concentration region, which precludes major structural changes and indicates that the unimer present at low concentrations also persists as a unique species at concentrations exceeding the bulk CMC. The $\mu^{\mathrm{ex}}$ of the A32B32 diblock copolymers submitted to the pore of D=30 monotonically decreases with C as well but becomes considerably lower than that of the copolymers in the pore of D=20 when $C>3\times 10^{-3}$ because of the formation of aggregates with low association numbers inside the pore, demonstrated by the $n_{\mathrm{free}}$ vs. C and $\nu_{\mathrm{free}}$ vs. C curves for D=30 in Figure 3b and Figure 4b, respectively. As the aggregates are small and the association/dissociation is reversible, we can still obtain the equilibrium data such as the partition coefficients, K, from the pore of D=30 (the orange curve in Figure 2a). However, the shapes of curves for the bulk and for the wide pores of D≥40, and their $\mu^{\mathrm{ex}}$ values, significantly differ from those for the relatively narrow pores of D=20 and 30. The initial parts of $\mu^{\mathrm{ex}}$ vs. C curves for the bulk and the wide pores of D≥40 at low C are similar to those for D≤30, but later the curves exhibit distinct break points, indicating a significant structural change. The breaks occur approximately at the bulk CMC, and then all $\mu^{\mathrm{ex}}$ decrease appreciably. In spite of the trivial differences in $\mu^{\mathrm{ex}}$ between the bulk and the pores, all the values of $\mu^{\mathrm{ex}}$ at C>CMC are considerably low and similar, indicating that the highly stable micelles do not form only in the bulk but can also form in the pores at C>CMC under certain conditions, i.e., if they cannot escape in bulk. Moreover, the strong association of copolymers in the wide pores results in the non-equilibrium frozen states, and consequently the partition coefficients obtained from the simulations starting from the pore are unreliably high as shown in Figure 4c. In summary, we carefully investigated the phase equilibria of amphiphilic A32B32 diblock copolymers partitioning between the bulk and the pores with inert walls and have shown the impact of the association of copolymers on the partition coefficient, K.

In the next part, we investigate the potential of the SEC characterization of AB diblock copolymers differing in the lengths of two blocks (the total length is constant, i.e., $N_{A}+N_{B}=64$). We simulated their partitioning between the bulk solvent and the pore of D=20 and varied the temperature, $T/T_{0}$, from 1.5 to 1.8. The dependences of K on C obtained from the simulations are shown in Figure 6. They are qualitatively similar to those shown in the previous figures and agree with the generally known effect of the solubility of block copolymers on their associating behavior. The solubility of copolymers decreases with the increasing relative length of the insoluble block and with decreasing temperature, and consequently their association tendency increases, which translates into variations of simulated K vs. C curves. The CMC values estimated from the simulations shift to lower C with the increase in length of the insoluble block, and simultaneously the K-drop becomes steeper and deeper (K approaches almost zero) as a result of the increasing association tendency. As the solubility of copolymers with long soluble blocks A is sufficiently high, these copolymers do not associate at all, which is shown by more or less constant or slightly increasing K vs. C plots, i.e., at $T/T_{0}=1.7$ and 1.8, indicating that the non-associated chains behave as effectively at elevated temperatures as polymers in good solvents [36,37].

At first glance, the effect of the relative length of blocks on K in the region of low concentrations can be surprising. K increases appreciably with the length of the insoluble block and with decreasing temperature. At $T/T_{0}=1.5$ (Figure 6a), the partition coefficients of A24B40 (green crosses) and A28B36 (orange squares) are even higher than 1 when C→0. This is not the simulation artifact, even though such values are not achievable in real SEC experiments. The increase of the partition coefficient, K, with the decreasing solubility of the copolymers and the extraordinary K>1 for copolymers with long solvophobic blocks can be explained relatively easily: they stem from an extremely strong unfavorable interaction between the solvent and block B. As the chains do not associate below the CMC and the collapse of the insoluble block does not sufficiently prevent the contacts of B segments with the solvent, the high-energy system of dissolved chains exploits another possibility to minimize the Gibbs function. The behavior is reminiscent of that of surface-active compounds that accumulate at interfaces [82,83]. The minimization of the number of unfavorable interactions is achieved when blocks B concentrate close to the inert pore wall. To reach this goal, the enthalpic driving force pushes the copolymer chains into the pore, where they acquire suitable conformations close to the wall.

In summary, Figure 6 depicts the changes in phase equilibria of dissolved associating systems during their transition from a selective solvent to a poor common solvent, later to the θ-solvent, and finally to a good common solvent with the increase in the ratio of solvophilic–to-solvophobic blocks and with temperature. We believe that the interesting outlined trends can be efficiently studied experimentally by temperature-modulated chromatographic techniques, e.g., by temperature–gradient interaction chromatography (TGIC) [3].

At the end of this article, we focus on the partitioning of A32B32 copolymers between the bulk phase and pores with adsorbing walls, which relates to the interaction chromatography (IC) mode. We assume that the pore wall is inert for the solvophilic block (A), i.e., $\epsilon_{\mathrm{WA}}=0$, and attracts the solvophobic block (B), i.e., $\epsilon_{\mathrm{WB}}<0$. One can imagine that if the attraction is very strong, the micelles in bulk will not form because all the chains will tenaciously adsorb on the pore wall, and K will be very high. However, this extreme case is unsuitable for real chromatography. Hence, the $\epsilon_{\mathrm{WB}}$ used is up to −0.9 and the temperature $T/T_{0}=1.7$, which is relevant for practical LC experiments. We performed two sets of simulations, which started with all chains either in the bulk or in the pore. In Figure 7a,b, we show the K vs. C curves obtained by simulations starting from the bulk phase for D=15 and D=60, respectively. For the reader’s convenience, the curves for $\epsilon_{\mathrm{WB}}=0.0$ (SEC mode) are also included. The comparisons between two groups of simulations with different initial conditions for D=15 and 60 are presented in Appendix A. As shown in Appendix A, the values of K for D=15 are identical because very few chains can aggregate in the narrow pore. However, for the wide pore of D=60, the data from the two sets of simulations differ significantly when $C>4.5\times 10^{-3}$ (Appendix A). The extremely high values of K provided by the simulations starting with all chains in the pore imply that the micellization of A32B32 copolymers takes place inside the pore and combines with the adsorption of chains on the pore wall. The micelles become trapped in pores similarly to the previously studied SEC systems (see Figure 4c). The reasons are similar in both cases, but in the IC regime, the escape of polymer chains from pores is even less probable because $K_{p}>1$ and the concentration of free chains in bulk is lower than that in pores, i.e., in this particular case lower than CMC, which precludes the association of chains in the bulk and thus eliminates the process that could drive the chains from the pores into the bulk phase. Even though K does not describe the equilibrium situation and the data in Appendix A are irrelevant for common IC chromatography, the obtained pieces of information are still useful for analyzing specific LC experiments, e.g., for the gradient or barrier chromatography [7]. In agreement with common LC processes, here we focus only on the case that all copolymers are initially in the bulk mobile phase (Figure 7a,b). Generally, K increases with increasing $|\epsilon_{\mathrm{WB}}|$ in the full range of studied concentrations for both D=15 and 30, and the increase in K is more pronounced in the dilute regime because the self-assembly in bulk dominates the partitioning at high concentrations, i.e., above the CMC. Nonetheless, Figure 7a demonstrates that the copolymers still undergo SEC separations in narrow pores, i.e., the maximum K for $\epsilon_{\mathrm{WB}}=-0.9$ is merely 0.28 (considerably lower than 1) because the attractive interactions of insoluble blocks with the pore wall are insufficient to offset the entropy loss caused by the severe confinement and are undoubtedly incomparable with the strong association in bulk. However, for the wide pore of D=60 with the strongest adsorption of $\epsilon_{\mathrm{WB}}=-0.9$ (see the magenta curve with inverted triangles in Figure 7b), K is approximately 1.8 when C→0, first increases with C, and reaches a maximum of K=2.5, indicating that the separation in the dilute regime obeys the IC mechanism and the adsorptive interaction prevails over the entropy loss.

We assume that the pronounced maximum and the shift of the fairly steeply decreasing part of K vs. C on the magenta curve (for the strong attractive interaction of insoluble beads with the walls of the wide pore) to higher concentrations followed by a slower but still appreciable K-decrease reflect the adsorption of insoluble blocks and consequent gradual saturation of the pore surface, followed by potential changes in the self-organization of chains due to the increase in C, e.g., by the formation of core-shell associates inside the pore. To prove this assumption, we plot further the concentration profiles of beads and present typical simulation snapshots.

Using the simulation data on the copolymer partitioning, we plot the radial concentration profiles of beads A and B, i.e., $\Phi_{A}$ vs. r and $\Phi_{B}$ vs. r, in the pore of D=60 at the high total concentration $C=8.9\times 10^{-3}$ for various $\epsilon_{\mathrm{WB}}$ in Figure 7c,d, respectively. Note that in contrast to the radial concentration profiles plotted in Figure 5b,c, the total number of beads in the pore (at constant total concentration, C) is not constant, and it increases with $|\epsilon_{\mathrm{WB}}|$, which slightly affects the absolute values of $\Phi_{A}$ and $\Phi_{B}$ but does not change the general trends. As expected, the concentration of solvophobic segments (B) near the inner surface of the pore is higher than that of solvophilic beads (A), and the difference increases with increasing $|\epsilon_{\mathrm{WB}}|$. The concentration profiles of beads A and B for $\epsilon_{\mathrm{WB}}=-0.9$ reveal that the solvophobic blocks are enriched in the layer near the pore wall, while the solvophilic ones favor the central region, which is also confirmed by the snapshot of the stationary phase in Figure 7e.

As the micellization of block copolymers and surfactants obeys the same principles, our computer study (particularly the simulations for the wide adsorptive pore, i.e., D=60 and $\epsilon_{\mathrm{WB}}=-0.9$) represents the first step towards the understanding of complex phase equilibria of surfactants in MLC. The structure of the micelles composed of dense solvophobic cores and less dense solvophilic shells is suitable for the solubilization or for the specific non-covalent binding of various molecules: (i) both the shells of surfactant micelles in bulk (mobile phase) and the soluble parts of unimers firmly adsorbed on pore walls can specifically interact with various soluble compounds, facilitate their penetration into pores, and control their chromatographic separation from non-interacting species [8]; and (ii) the cores of micelles in bulk can solubilize the otherwise insoluble compounds, and the insoluble parts of adsorbed unimers in pores can interact favorably with them, which can intermediate their partitioning, transport, and separation in the column.

We envisage that our study of the partitioning of copolymers and surfactants between the bulk and the wide pore, of which the inner surface strongly adsorbs solvophobic units (e.g., D=60 and $\epsilon_{\mathrm{WB}}=-0.9$), demarks the correct approach towards the investigation of the MLC separation mechanism. The synergy of the micellization in the bulk, the adsorption of surfactants on the pore wall, and the attractive interaction with analytes enables the versatile tuning of MLC elution, and the present study already reveals some principles of this complex molecular mechanism. The computational investigation of the MLC process based on this model will be presented in our future work.

## 4. Conclusions

The performed Monte Carlo simulations investigate the intriguing concentration-dependent partitioning of amphiphilic diblock copolymers dissolved in selective solvents between the bulk phase and the pore, reveal its most important trends, elucidate its molecular mechanism, and outline the impact of complex phase equilibria on the retention behavior of associating polymers in LC columns. Our simulations show a unique concentration dependence of the partition coefficient, K, in the SEC regime, which exhibits a sharp decrease in the CMC region as a result of the copolymer self-assembly and is different from those for non-associating polymers at good, poor, and θ-solvent conditions [34,35,36,37]. Above the CMC, most of the chains aggregate in the mobile phase, and the concentration of unimers (non-associated chains) remains low, constant, and equal to the CMC [71]; therefore, only a few chains enter the pores, and the partition coefficient is much lower than that of the non-associated chains at C<CMC.

Our simulation data agree with the results of other authors [40,46,48] and confirm that large aggregates can be formed in wider pores, e.g., in the cylindrical pore of D=60 if the chains are enclosed in the pores and cannot escape in the bulk, but they preclude (i) the spontaneous transport of large micelles from the bulk to the pores and (ii) the micellization in the pores if the chains in the two phases communicate on the basis of reversible phase equilibria. The results prove that the system obeys basic assumptions formulated in our working hypothesis (Equations (Equation 1) and (Equation 2)). Nevertheless, in agreement with our restraint concerning the micellization inside the pores, the study shows that the association process described by Equation (Equation 3) does not take place in studied systems.

As expected, the characteristics describing the partitioning of copolymers differing in composition, i.e., the CMC and the steepness of the drop in the K vs. C curve, depend on the length of the blocks and on the temperature, thus reflecting the solubility and association tendency of these copolymers. This observation suggests the possibility of discerning of copolymers differing in composition by SEC and IC.

Finally, we simulated the partitioning of amphiphilic diblock polymers between the bulk and strongly adsorbing pores (attracting only the insoluble blocks). When the concentration is higher than the CMC, the chains in the bulk form large micelles with solvophobic cores, and the solvophobic blocks of non-associated chains in cylindrical pores are tenaciously adsorbed on the inner surface, whereas the soluble blocks concentrate near the central region of the pore mimicking the flexible ligands or tethered chains [84]. This implies that various soluble molecules could be solubilized both in micellar shells in the mobile phase and in the concave brushes of soluble blocks in the stationary phase, and the chromatographic behavior of these analytes can be tuned and optimized by the choice of the micelle-forming surfactants or polymers and by adjusting the interactions, e.g., by changing the temperature and solvent. In summary, our simulations under the IC condition indicate a direction towards a deeper understanding of the separation mechanism of micelle liquid chromatography (MLC), which is a promising green analytical method and exhibits great application potential in the separation and analysis of biomolecules [9,10,11,12,13,14,15].

## Figures and Tables

**Figure 1 polymers-14-03182-f001:**
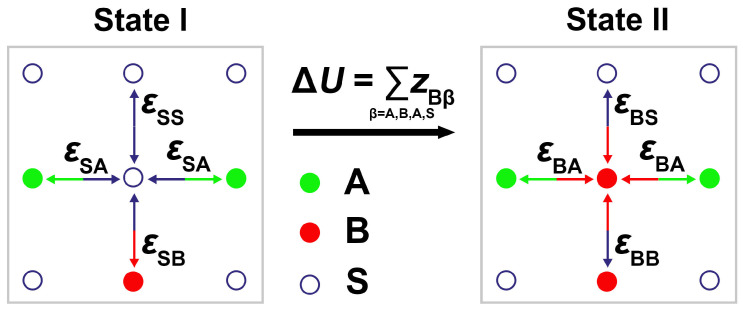
2D illustration of the energy contribution due to the exchange of a solvent bead S and polymer bead B in a given lattice position. In State I, the central lattice point is occupied by bead S, and its contribution to the total energy of system is $U_{I}=\varepsilon_{\mathrm{SA}}+\varepsilon_{\mathrm{SB}}+\varepsilon_{\mathrm{SA}}+\varepsilon_{\mathrm{SS}}$. In State II, this point contains bead B, and its contribution to the total energy of system is $U_{\mathrm{II}}=\varepsilon_{\mathrm{BA}}+\varepsilon_{\mathrm{BB}}+\varepsilon_{\mathrm{BA}}+\varepsilon_{\mathrm{BS}}$. Therefore, the energy change due to the replacement of the solvent bead by polymer bead is $U=U_{\mathrm{II}}-U_{I}=2\left(\varepsilon_{\mathrm{BA}}-\varepsilon_{\mathrm{SA}}\right)+\left(\varepsilon_{\mathrm{BB}}-\varepsilon_{\mathrm{SB}}\right)+\left(\varepsilon_{\mathrm{BS}}-\varepsilon_{\mathrm{SS}}\right)=2z_{\mathrm{BA}}+z_{\mathrm{BB}}+z_{\mathrm{BS}}$.

**Figure 2 polymers-14-03182-f002:**
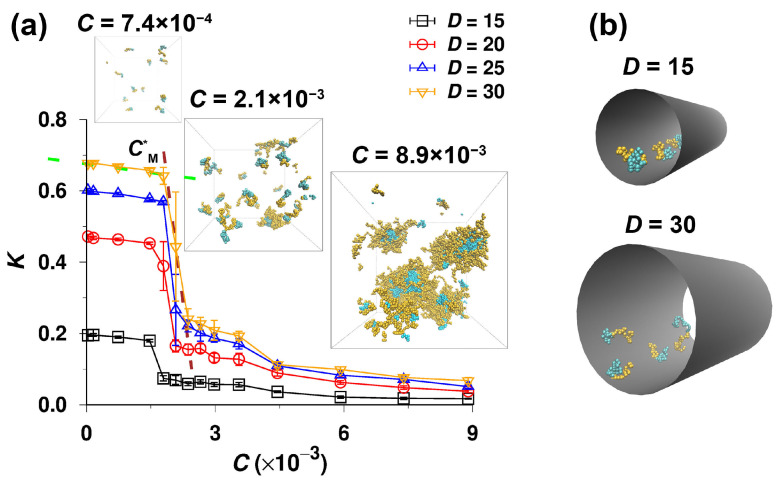
(**a**) Partition coefficient, K, of A32B32 copolymers partitioning between the bulk and the pore, as a function of total concentration of beads, C, for various pore diameters, D. Typical snapshots of copolymers in the bulk corresponding to the partitioning (D=30) at the low ($C=7.4\times 10^{-4}$), medium ($C=2.1\times 10^{-3}$) and high ($C=8.9\times 10^{-3}$) concentrations are also given, where the orange and cyan beads represent the solvophilic (A) and solvophobic (B) segments, respectively. (**b**) Typical snapshots of copolymers in the cylindrical pores of D=15 and 30. The concentrations of chains in the pores correspond to the partitioning between two phases at $C=8.9\times 10^{-3}$. All snapshots were rendered with VMD [70] .

**Figure 3 polymers-14-03182-f003:**
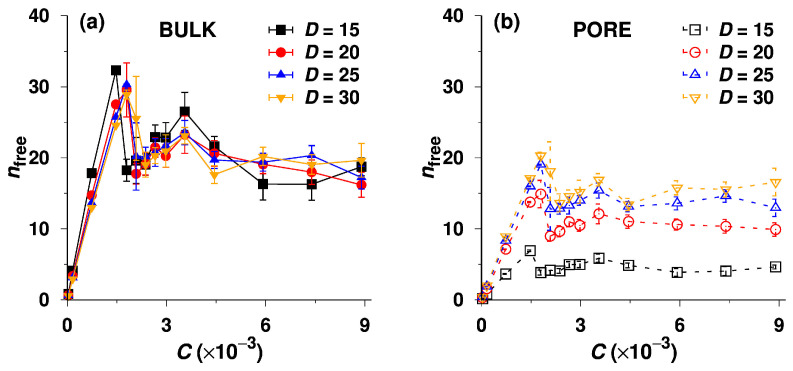
(**a**) Number of free chains, $n_{\mathrm{free}}$, in the bulk phase as a function of the total concentration, C, for various pore diameters, D. (**b**) Number of free chains, $n_{\mathrm{free}}$, in the pore as a function of C, for various D.

**Figure 4 polymers-14-03182-f004:**
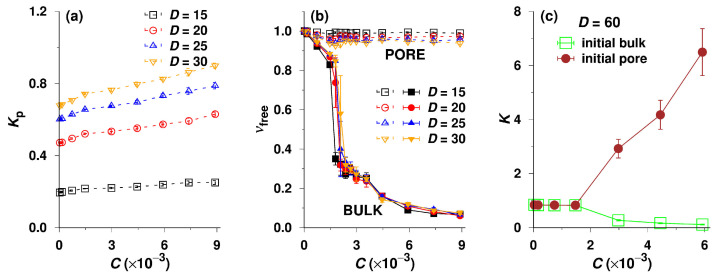
(**a**) Partition coefficient of non-associated A32B32 copolymers (unimers), $K_{p}$, as a function of total concentration, C, for various pore diameters, i.e., D=15, 20, 25 and 30. (**b**) Fraction of free chains, $\nu_{\mathrm{free}}$, in both the bulk and the pore as a function of C for various D. (**c**) The effective partition coefficient, K, of A32B32 copolymers as a function of C for the wide pore of D=60. The legends “init bulk” and “init pore” imply the two groups of simulations initially starting from the bulk and the pore, respectively.

**Figure 5 polymers-14-03182-f005:**
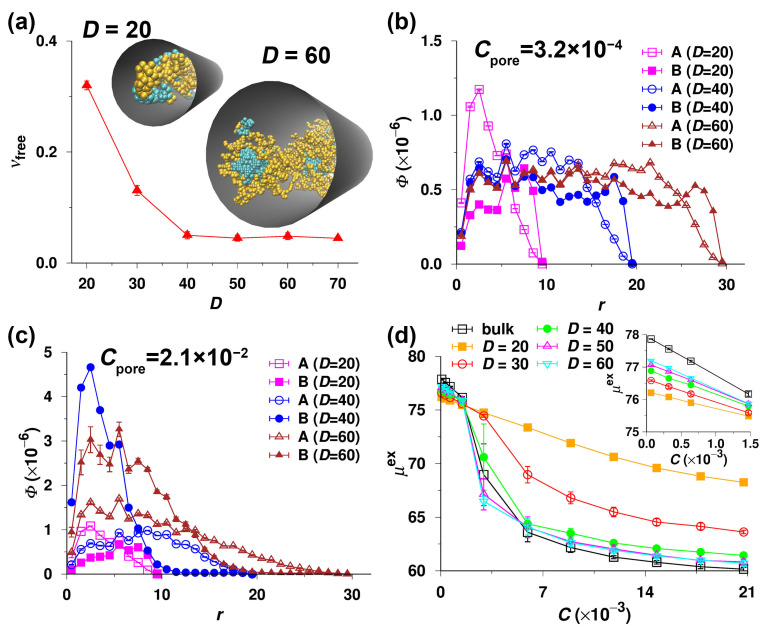
(**a**) The fraction of free chains, $\nu_{\mathrm{free}}$, in the pore as a function of the pore diameter, D. The concentration $C_{\mathrm{pore}}=2.1\times 10^{-2}$ for all D. The typical snapshots of A32B32 diblock copolymers in a selective solvent confined in the pores of D=20 and 60 are embedded, where the orange and cyan beads represent the solvophilic (A) and solvophobic (B) segments, respectively. (**b**) The concentration of beads, Φ, as a function of the distance to the pore center in the plane perpendicular to the pore axis, r, for $C_{\mathrm{pore}}=3.2\times 10^{-4}$ and various D. (**c**) Φ vs. r for $C_{\mathrm{pore}}=2.1\times 10^{-2}$ and various D. (**d**) Excess chemical potential, $\mu^{\mathrm{ex}}$, as a function of concentration, C, for the A32B32 diblock copolymers in the bulk and in the pores of various D. The zoomed-in view of $\mu^{\mathrm{ex}}$ vs. C at $C\leq 1.5\times 10^{-3}$ is shown in the inset. Note that all the data shown in panels a to d were obtained from the simulations which were individually performed in the bulk and in the pores.

**Figure 6 polymers-14-03182-f006:**
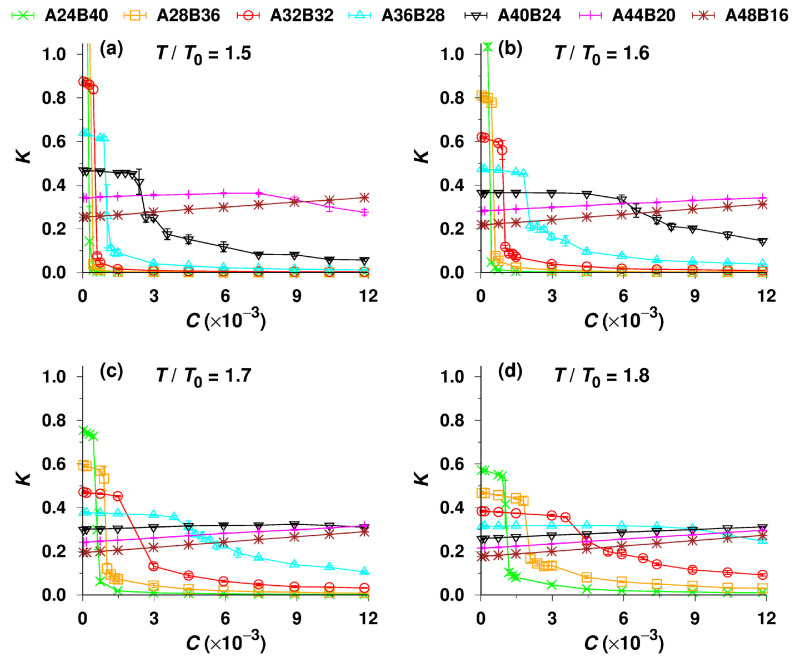
Partition coefficient, K, of seven different AB diblock copolymers as a function of total concentration, C, for the temperatures $T/T_{0}=1.5$ (**a**), 1.6 (**b**), 1.7 (**c**), and 1.8 (**d**). The pore diameter D=20, and the inner surface is inert, i.e., $\epsilon_{\mathrm{WA}}=\epsilon_{\mathrm{WB}}=\epsilon_{\mathrm{WS}}=0$.

**Figure 7 polymers-14-03182-f007:**
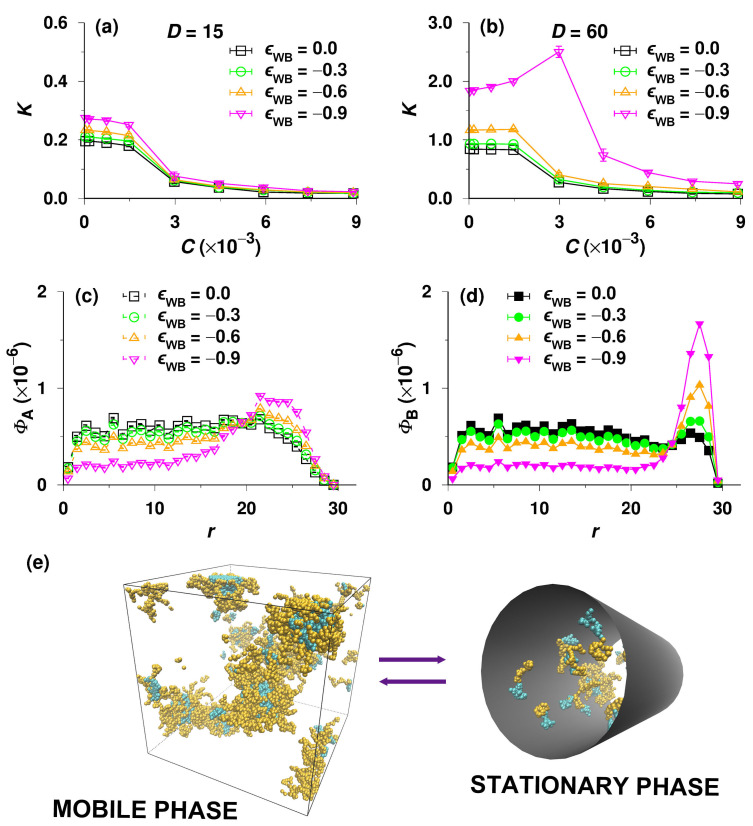
(**a**) Partition coefficient, K, of A32B32 copolymers as a function of total concentration, C, for various adsorption strengths, $\epsilon_{\mathrm{WB}}$. The diameter of cylindrical pore D=15. (**b**) K vs. C for various $\epsilon_{\mathrm{WB}}$ for D=60. (**c**,**d**) The concentrations of beads A and B, $\Phi_{A}$ and $\Phi_{B}$, in the cylindrical pore of D=60 as functions of the distance to the pore center in the plane perpendicular to the pore axis, r, for various $\epsilon_{\mathrm{WB}}$. (**e**) Snapshots of both the mobile phase (bulk) and the stationary phase (pore) for $C=8.9\times 10^{-3}$, D=60, and $\epsilon_{\mathrm{WB}}=-0.9$.

## Data Availability

The data presented in this study are available upon request from the corresponding authors.

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
