# Peer review of "Modeling the Phase Equilibria of Associating Polymers in Porous Media with Respect to Chromatographic Applications"

_polymers, 2022, doi:10.3390/polym14153182_

Round 1

Reviewer 1 Report

The primary value of the work appears to be in simulating a qualitative description of the processes that will go on in an associating system with pores.  The parameters used are somewhat arbitrary and there is little description of the impact of the particular values chosen.

The second apparent drop in K with concentration (e.g. Fig 2 D=30 C=3.5 - 4.5) is not adequately described.

The authors should avoid relying on reference to the colour of the line on a graph when discussing results - refer to the parameter of the dataset.

Although the drop in K is qualitatively linked to the CMC and the formation of micelles, this is not independently measured beyond consideration of the fraction of free chains.  Some assessment of the micelle size/number of chains involved in each cluster would improve the clarity here.  This assessment of the formation of micelles could then be compared with the case in which no pores are present.

The conclusions section should be more specific about what can be learned from this study with regard to the behaviours.  The 'narrative' of the paper is at times difficult to follow.  The value of the paper will be to leave the reader with a good 'picture' of the kind of processes that can happen under different circumstances and this is what should be summarised clearly in the conclusions, which at present puts too high an emphasis on what the modelling approach may achieve in future.

Reviewer 2 Report

Comments for authors on manuscript polymers-1826431: "Modeling the Phase Equilibria of Associating Polymers in Porous Media with Respect to Chromatographic Applications” by Xiu Wang , Zuzana Limpouchová, Karel Procházka, Rahul Kumar Raya , Yonggang Min.

 In this manuscript the authors describe a complete computational study with the aim of elucidate the mechanism of the amphiphilic diblock copolymers chromatographic behavior. These studies provide useful hints for the analysis of experimental elution curves of associating polymers. This report was carefully performed and included several interesting observations, a significant advance could find from present report. The design of models that can predict the behaviour of macromolecules is of interest to the scientific community and to industry. Therefore, the manuscript is suitable for publication in Polymers in the present form.

- In my opinion the authors should apply these theoretical models under experimental conditions with real polymers and in different solvents, that is, go one step further, not just stick to theoretical predictions.
